# College Students’ Degree of Support for Online Learning during the COVID-19 Pandemic and Associated Factors: A Cross-Sectional Study

**DOI:** 10.3390/ijerph192416814

**Published:** 2022-12-14

**Authors:** Xincheng Huang, Yuqian Deng, Pu Ge, Xiaonan Sun, Mengjie Huang, Hejie Chen, Yanyan Wang, Baojun Suo, Zhiqiang Song, Yibo Wu

**Affiliations:** 1School of Economics and Management, Beijing Institute of Graphic Communication, Beijing 102600, China; 2Xiangya School of Nursing, Central South University, Changsha 410008, China; 3Institute of Chinese Medical Sciences, University of Macau, Macao 999078, China; 4Department of Social Science and Humanities, Harbin Medical University, Harbin 150081, China; 5School of Public Health, Shandong University, Jinan 250012, China; 6Department of Gastroenterology, Peking University Third Hospital, Beijing 100191, China; 7School of Public Health, Peking University, Beijing 100871, China

**Keywords:** family communication, online learning, social support, negative status, college students, COVID-19, SEM, coronavirus disease 2019

## Abstract

Background: Educational institutions worldwide have experienced the suspension of offline teaching activities in favor of online teaching due to the outbreak of the COVID-19 pandemic. However, few studies have focused on the degree of support for online learning among college students in mainland China. Therefore, the aim of this study was to investigate the degree of support for online learning among Chinese college students during the epidemic and whether depression, loneliness, family communication, and social support were associated factors. Methods: A questionnaire was used to collect cross-sectional data from 9319 college students in mainland China, and a structural equation model was analyzed. Results: The results of the study showed high degrees of support for online learning among Chinese college students during the COVID-19 pandemic, with more than half expressing support. The SEM (Structural Equation Modeling) results showed that depression had a negative and significant effect on college students’ support for online learning (β = −0.07; *p* < 0.001); family communication had a positive and significant effect on college students’ support for online learning (β = 0.09; *p* < 0.001); social support had a positive and significant effect on college students’ support for online learning (β = 0.11; *p* < 0.001). Conclusions: Social support and family communication can alleviate the negative psychological status of college students, and depression plays a mediating role in the effect of social support and family communication on college students’ degree of support for online learning. In addition, a significant chain-mediating effect was found of family communication, loneliness, and depression between social support and college students’ degree of support for online learning. Government and education institutions must focus on college students’ mental health issues and consider family interventions and general support that college students require.

## 1. Introduction

Since the outbreak of Coronavirus Disease 2019 (COVID-19), educational institutes have been affected significantly. The United Nations Children’s Fund (UNICEF) report in 2021 found that schools of more than 168 million children globally had been closed for almost a year. In China, on 6 February 2020, the Ministry of Education took the step of “Suspending Classes Without Stopping Learning” to ensure that students could continuously study with online platforms at home [1]. Most universities adopted an online learning policy to react to the COVID-19 pandemic. Traditional offline learning has been forced to transform into online learning to provide ongoing education to students [2]. As defined as access to learning resources through some technological means, online learning is the recent version of distance learning and e-learning, emphasizing access to educational opportunities [3]. The digital transformation of educational institutions has a long history, especially during the COVID-19 pandemic, and online learning is increasingly becoming a topic of concern [4]. By 2022, college students in multiple provinces will still be experiencing online learning, and more will be doing so due to recurring outbreaks and China’s strict prevention and control policies. However, for college students, online learning has some problems, such as stress, difficulties completing school work [5], and information overload in an online learning context [6]. In other countries, many students reported that online learning made them unsatisfactory and had a negative attitude toward online learning [5]. A study from the Philippines showed that college students face variable challenges in online learning, while the largest one is linked to their learning environment at home, which may have a great impact on the quality of the learning experience and students’ mental health [7]. On the contrary, another research study showed a higher interest in online learning among college students, which is consistent with a study in Jordan [8,9]. Therefore, understanding students’ degree of support for online learning (DSOL) during the COVID-19 pandemic and the factors influencing their engagements becomes imperative.

Attitude theory suggests that attitudes are individuals’ positive or negative evaluative responses to a person or things, usually rooted in beliefs and forming specific psychological dispositions [10]. The degree of support is essentially an expression of attitude, and college students’ DSOL indicates their attitude toward online learning. According to previous studies, attitudes are influenced by various factors, which can usually be divided into three aspects: personal psychological level, family environment level, and social environment level [11,12,13].

First, the degree of support is an attitude that is easily influenced by the individual inner and psychological level. Many kinds of research have shown that the COVID-19 pandemic significantly impacted college students’ psychology from China or other countries [14,15,16,17]. As a reaction to the pandemic, college students generally have psychological problems such as depression and loneliness, which eventually lead to a negative online learning experience [18]. On the one hand, it is interesting to note that most research about the relationship between college students’ performances in school and negative moods has taken place in traditional and in-person courses [19]. However, depression can also affect students’ learning experiences online. Depression can negatively affect their creativity, concentration, and motivation to learn, which is desperately needed for online learning [20,21]. On the other hand, due to the impact of epidemic prevention and control, some college students were forced to study online at home, and loneliness increased because their relationships with others were reduced [22,23]. Theoretically, online learning may lead to loneliness, making college students more opposed to online learning. However, few studies have examined the relationship between college students’ loneliness and online learning. Therefore, it is worthwhile to investigate whether depression and loneliness among Chinese college students during the COVID-19 pandemic negatively impact their DSOL. Based on the above, we proposed the following research hypotheses:

**Hypothesis** **1a.**
*Depression has a significantly negative effect on DSOL.*


**Hypothesis** **1b.**
*Loneliness has a significantly negative effect on DSOL.*


Secondly, the family environment is also a critical level that influences college students’ DSOL, and families influence students’ learning in several ways. Generally, the poorer the family infrastructure, the poorer the student’s performance, parental education, literacy, the family’s socioeconomic status, and the extent of parental involvement in the child’s learning, which all influence their attitudes and academic performance [24]. The online learning environment is mainly at home, blurring the line between home and learning environments. College students have to communicate with family members more often, leading to conflicts [25], which means that a better family communication environment is needed to reduce conflict. According to Family System Theory, family members influence each other, and the support and conflict between family members affect students’ learning [26]. Several studies have investigated the relationship between family and students’ attitudes toward online learning [27,28,29]. For example, a study of middle school students found that the relationship with family members is an influential variable impacting students’ attitudes about online learning during the COVID-19 pandemic [30]. However, family communication as an important factor has rarely been included in studies, especially in the background of strict prevention and control of the pandemic in China. Therefore, family communication is used in this paper to explore the relationship between family and college students’ attitudes toward online learning. Based on the above, we proposed the following research hypotheses:

**Hypothesis** **2.**
*Family communication has a significantly positive effect on DSOL.*


Thirdly, the social environment is another important external factor influencing college students’ DSOL. Social support is the support and assistance individuals obtain from social groups through interactions with others in their social relationships (including family, friends, teachers, colleagues, and the corresponding social security departments) [31]. Social support can influence people’s satisfaction, attitudes, and psychological statuses [32,33]. Some studies have shown that college students face many challenges in online learning, which can negatively affect their attitudes toward online learning [34]. In such circumstances, college students often need support to improve their studies [35]. In a sense, social support measures the extent to which help from others is possible and the extent to which it constitutes a supportive network [36]. In general, the more social support students perceive, the better they will be at addressing the problems they encounter in the learning process and the better able they will be at overcoming the challenges posed by their objective environment [37]. During the COVID-19 lockdown, social support is also the predictive factor influencing college students’ online learning engagement [38]. Therefore, we proposed the following research hypothesis:

**Hypothesis** **3.**
*Social support has a significantly positive effect on DSOL.*


Finally, according to the stress-buffering model [39], social support is proven to buffer an individual’s depression and loneliness [40]. Moreover, the relationship between social support and mental health has been confirmed by many research findings [41]. For example, a study conducted in Shaanxi province, China, revealed that social support is an essential factor affecting college students’ mental condition; a similar conclusion was also found in a study from Iran [42,43]. Social support is one of the important external resources to buffer stressors, and even if individuals are not facing any stressful events, their mental health will benefit to some extent from the perceived social support [44]. In the background of the COVID-19 pandemic, college students face deepening stress from the academic and family levels, leading to a negative psychological state [45], and social support serves as an alleviating factor. Family System Theory suggests that family members can influence family members’ emotions, attitudes, and behaviors. The status of family communication can directly affect the adolescent’s mental health [46]. Communication between parents and their children, especially in academic affairs, can easily lead to conflicts and negative psychological emotions on both sides [47]. Especially during the pandemic, college students’ online learning environment is mainly at home, and the influence of family on college students’ attitudes is prominent. In addition, many kinds of research have revealed the relationship between social support and family functioning. For children and adolescents with illnesses or women in the perinatal period, support from family, friends, or other social relationships can greatly improve their family relationships [48], enhance their family cohesion [49], and have a greater impact on family functioning [50]. Family communication is an important aspect of family functioning. Therefore, it is reasonable to believe that social support will have the same effect on family communication. Thus, we proposed the following research hypotheses:

**Hypothesis** **4a.**
*Social support has a significantly negative effect on depression.*


**Hypothesis** **4b.**
*Social support has a significantly negative effect on loneliness.*


**Hypothesis** **5a.**
*Family communication has a significantly negative effect on depression.*


**Hypothesis** **5b.**
*Family communication has a significantly negative effect on loneliness.*


**Hypothesis** **6.**
*Social support has a significantly positive effect on family communication.*


The study’s primary objective was to investigate the DSOL among college students and the factors influencing it in the background of the COVID-19 pandemic. The secondary objective was to explore the associations between negative mental status (depression and loneliness) and DSOL. The third objective was to test the role of social support and family communication in alleviating college students’ poor mental status, which affected their DSOL. Then, the potential mediation between social support and DSOL via depression, loneliness, and family communication was investigated. Based on the above discussion, the model hypothesis graph for this study was constructed and is presented in Figure 1.

## 2. Materials and Methods

### 2.1. Participants and Procedures

A multi-stage sampling method was adapted from 20 June to 31 August in the “Psychology and Behavior Investigation of Chinese Residents in 2022” [51]. Based on the Chinese population pyramid, residents in a total of 148 cities, 202 districts and counties, 390 townships/towns/sub-districts, and 780 communities/villages (excluding Hong Kong, Macao, and Taiwan) from 23 provinces, 5 autonomous regions, and 4 municipalities directly under the central government in China were selected in this study [52]. At least one investigator or one investigation team was recruited in each city, each investigator was responsible for collecting 30–90 questionnaires, and each team was responsible for collecting 100–200 questionnaires. The questionnaire was distributed one-on-one and face-to-face to the public by trained investigators. A total of 31,480 questionnaires were distributed, and 30,505 valid questionnaires were finally collected, with an effective rate of 96.9%, as shown in Figure 2. This study has been officially registered in the China Clinical Trial Registry (Registration No.: ChiCTR2200061046). All methods were performed in accordance with relevant guidelines and regulations. In this study, we selected participants who were college students. 

Strict inclusion and exclusion criteria were developed to screen the participants. Inclusion criteria: (1) college students, including specialists, undergraduates, and graduate students; (2) had the nationality of the People’s Republic of China (excluding Hong Kong, Macau, and Taiwan); (3) China’s permanent resident population with an annual travel time ≤ 1 month; (4) participated in the study and filled in the informed consent form voluntarily; (5) participants can complete the questionnaire survey by themselves or with the help of investigators; (6) participants can understand the meaning of each item in the questionnaire. The exclusion criteria were: (1) persons with unconsciousness or mental disorders; (2) those participating in other similar research projects. After excluding invalid questionnaires: (1) filling time ≤ 100 s; (2) logically inconsistent; (3) incompletely filled. 9319 college students were enrolled in this study, and the sample number met the minimum requirements set by Bentler and Chou [53]. Figure 2 shows a detailed flowchart of the enrollment.

### 2.2. Measures

#### 2.2.1. Sociodemographic Characteristics

Information about age, sex, location of residence, and monthly per capita family income was collected. 

#### 2.2.2. Degree of Support for Online Learning

Participants were asked, “How well do you support the implementation of online learning during the pandemic of COVID-19?” (on a scale ranging from 1 to 100 points: 1–33 = not supportive; 34–66 = general; 67–100 = supportive). 

#### 2.2.3. Self-Reported Quarantine Status

Participants were asked three questions in the questionnaire: (1) Are you currently quarantined from home? (2) Is your city under closed management? (3) Is your community under closed management? The options for these three questions are: 0 = No; 1 = Yes.

#### 2.2.4. Depression

Depression was assessed by the 9-item Patient Health Questionnaire (PHQ-9), validated in many previous studies [54,55]. It is a simple and validated self-rating scale for depressive disorders that can effectively screen individuals for depression and is widely used internationally. The scale consists of nine common depressive symptoms that participants rate based on their feelings. The questionnaire is scored on a four-point scale (0 = not at all to 3 = nearly every day in the past 2 weeks; range = 0–36, with higher scores indicating more severe depression). The Cronbach’s α of the PHQ-9 in this study was 0.914. Latent variables were created through three random parcels as manifest indicators. Details of this scale can be reviewed in Appendix A.

#### 2.2.5. Loneliness

Loneliness was measured using the Three-Item Loneliness Scale (T-ILS), which contains 3 questions (How often do you feel isolated from others? How often do you feel you lack companionship? How often do you feel left out?) [56]. In large higher education surveys, the Three-Item Loneliness Scale (T-ILS) is increasingly being used [56]. The scale is rated on a 3-point Likert scale (1 = hardly ever, 2 = sometimes, and 3 = often). The Cronbach’s α of the T-ILS in this study was 0.885. Details of this scale can be reviewed in Appendix A.

#### 2.2.6. Family Communication 

The Family Communication Scale (FCS) contains 10 items and is measured on a 5-point Likert scale ranging from “strongly disagree” to “strongly agree” [57]. The objective of the scale is to measure the quality of communication between family members regarding the exchange of ideas, information exchange, level of concern, openness, confidence, and emotions between family members. It values positive communication skills such as clear and congruent messages, empathy, supportive phrases, and effective problem-solving skills. The Cronbach’s α of FCS in this study was 0.970. Latent variables were created through three random parcels as the manifest indicators. Details of this scale can be reviewed in Appendix A.

#### 2.2.7. Social Support

Social support was measured by the Perceived Social Support Scale, which contains 12 items on a 7-point Likert scale ranging from “extremely disagree” to “extremely agree” [58]. The Perceived Social Support Scale (PSSS) is a social support scale that emphasizes self-understanding and self-perception. It measures the degree of support individuals perceive from various sources of social support, such as family, friends, and others. It reflects the total degree of social support perceived by individuals with a total score. The scale has been validated in several studies. The Cronbach’s α of PSSS in this study was 0.885. Latent variables were created through three random parcels as the manifest indicators. Details of this scale can be reviewed in Appendix A.

### 2.3. Statistical Analysis

Analyses were conducted using SPSS 25.0 and AMOS 25.0, and a two-sided *p* below 0.05 was considered statistically significant. The descriptive statistics of the sociodemographic characteristics (mean, standard deviation, and number/percentage) were calculated. A chi-square test was used to examine the differences among DSOL, gender, location of residence, and self-reported quarantined status. The Pearson correlation was used to analyze the correlations between the parameters and the predictive factors. SEM analysis with full information likelihood estimation was used to test the hypothesized mediation models. Testing for the direct, indirect, and total effects was based on 2000 bootstrapped samples; effect estimates and bias-corrected 95% confidence intervals (CI) were derived. Indices of good model fit included a root-mean-square error of approximation (RMSEA) < 0.06 and a comparative fit index (CFI) and Tucker Lewis Index (TLI) > 0.95 [59]. 

## 3. Results

### 3.1. Sample Characteristics

In the full sample, the mean and standard deviation of the age was 20.55 ± 4.02 years old; 40.2% were males; 50.4% were in town; nearly 90% of monthly per capita family income was under 9000 yuan; 3.4% were quarantined at home; 4.4% lived in the communities under closed management; 6.4% lived in the cities under closed management. Regarding DSOL, 14.1% were not supportive, 34.5% were general, and 51.4% were supportive. The average score for college students’ depression was 16.70 (SD = 5.74), the average score for loneliness was 4.97 (SD = 1.68), the average score for family communication was 36.60 (SD = 8.94), and the average score for social support was 14.95 (SD = 3.85). The results are presented in Table 1.

### 3.2. Common Method Bias Test

The measurement of subjects at the same time may lead to Common Method Bias in the data. Therefore, the Harman one-way ANOVA was used to test this issue in this study. The test results showed that the largest unrotated factor in this study explained only 38.61% (less than 50%) of the total variance. The results of this analysis indicated that the potential common method bias in the current sample data was within an acceptable range for in-depth empirical analysis.

### 3.3. Differences in DSOL among Age, Location of Residence, and Self-Reported Quarantined Status

DSOL was statistically significant with gender (*p* < 0.001), quarantined at home (*p* < 0.001), and city under closed management (*p* < 0.05). A significantly higher percentage of male college students, college students quarantined at home or in the city under closed management, chose not to support online learning. However, there was no significant difference in the DSOL among college students in different locations of residence. The results are presented in Table 2.

### 3.4. Associations between Social Support, Family Communication, Depression, Loneliness, and the Degree of Support of Online Learning

Social support exhibited moderate and positive correlations with family communication (r = 0.46, *p* < 0.01) and small and positive correlations with DSOL (r = 0.16, *p* < 0.01), but small and negative correlations with depression (r = −0.17, *p* < 0.01) and loneliness (r = −0.20, *p* < 0.01). Family communication exhibited small and negative correlations with depression (r = −0.21, *p* < 0.01) and loneliness (r = −0.27, *p* < 0.01), but small and positive correlations with DSOL (r = 0.18, *p* < 0.01). Depression exhibited moderate and positive correlations with loneliness (r = 0.58, *p* < 0.01) but small and negative correlations with DSOL. Finally, loneliness exhibited small and negative correlations with DSOL (r = −0.11, *p* < 0.01). The results are presented in Table 3.

### 3.5. Structural Equation Modeling 

The SEM models (CFI = 0.990, TLI = 0.987, RMSEA = 0.042, and SRMR = 0.021) achieved a good model fit. As shown in Figure 3 and Table 4, depression had a significant and negative effect on DSOL (*β* = −0.07; *p* < 0.001), but the effect size was small; the effect of loneliness on DSOL was insignificant. Social support (*β* = 0.09; *p* < 0.001) and family communication (*β* = 0.11; *p* < 0.001) had a significant and positive effect on DSOL with small effect sizes; social support had a significant and positive effect on family communication (*β* = 0.50; *p* < 0.001) with a middle effect size. Social support (*β* = −0.03, *p* < 0.05; *β* = −0.10, *p* < 0.001) and family communication (*β* = −0.03, *p* < 0.05; *β* = −0.21, *p* < 0.001) had a significant and positive effect on depression and loneliness, respectively. Loneliness (*β* = 0.64; *p* < 0.001) had a significant and negative effect on depression with a strong effect size. Finally, five indirect paths between Social Support and DSOL were significant, and the results are shown in Table 5.

## 4. Discussion

The study found high degrees of support for online learning among Chinese college students during the COVID-19 pandemic, with more than half expressing support. College students who are male or quarantined at home are more unsupportive of online learning than those who are not. College students living in the communities under management did not show significant differences in DSOL, while those living in cities showed extremely weak differences. In addition to loneliness, social support, family communication, and depression were all predictors of DSOL, but the effects were weak. Therefore, H1a, H2, and H3 were supported, but H1b was not. Both social support and family communication alleviated the depression and loneliness college students suffered during the COVID-19 pandemic, but the effects were also relatively weak. Thus, H4a, H4b, H5a, and H5b were supported. Finally, social support strongly affected family communication, which is in line with our research expectations; thus, H6 was supported. In terms of mediating effects, social support and family communication can indirectly affect DSOL via depression, or they can have a mediating chain effect first via loneliness and then via depression.

At the individual level, college students’ poor inner mental status (depression) negatively influenced college students’ DSOL. This finding is also consistent with an existing study, which explored the impact of depression on college students’ online learning during the lockdown [18]. The emergence of depression symptoms increases the mental pain of college students at different levels, causing them excessive stress, and the many inconveniences of online education can cause them to lose interest and confidence in online learning [60]. It is worth noting that the relationship between loneliness and college students’ DSOL is inconsistent with our study hypothesis. The inconsistency may be because this survey was conducted mainly in the summer months of July and August when college students are on vacation and do not feel as dependent on campus. In addition, communication with family members and psychological and material assistance from community members to college students will somewhat alleviate their loneliness, thus making them less resistant to online learning [61,62]. Communication with family and support from outside helps alleviate their feelings of loneliness. Nonetheless, we found that the loneliness of college students during the pandemic could largely explain their depressive symptoms, which is consistent with the findings of an existing study [63]. Therefore, online educational institutions must consider college students’ mental health status. Educational institutions can provide a good prerequisite for college students to actively participate in online learning by taking the initiative to communicate and understand the psychological problems of college students in the epidemic background, and cooperate with relevant medical institutions to help them alleviate their possible psychological barriers.

At the family level, family communication negatively affects college students’ depression and loneliness; family communication also positively influences college students’ DSOL. First, online learning activities during the pandemic were primarily conducted at home. Maintaining relationships with family members was essential, whose assistance was often needed to resolve learning difficulties (e.g., stress, difficulties, and information overload). This finding is consistent with existing studies. For example, a survey from Malaysia showed that 51.6% and 20.6% of students reported that family members harmed their online learning and that living with family members tended to present more challenges, which, in turn, affected the effectiveness of online learning [64]. Thus, college students with a better home communication environment may be more supportive of online learning. Secondly, according to Family System Theory, family communication is closely linked to family members’ emotional, physical, and psychological activities [65]. Good family communication can increase adolescents’ self-confidence in their bodies by alleviating depression [66,67], which may explain to some extent our findings. Family members can play an active role when college students show symptoms such as depression, and they can alleviate their negative feelings through communication and interactions. Especially in the epidemic background, family members’ communication is essential to alleviate college students’ negative psychology and enhance their confidence in their learning. In the adolescent population, family communication tends to alleviate their loneliness, especially among bullies [68], which aligns with our findings. Thirdly, consistent with the findings of existing studies, family communication can adjust college students’ behaviors and attitudes (DSOL) by alleviating negative psychological status. Finally, research has shown that men are more likely than women to have family conflicts [69], which may be why more male college students are unsupportive of online learning. In addition, college students quarantined at home may also be more susceptible to conflict because they need to communicate frequently with family members, thus resulting in more opposition to online learning than non-quarantined students. Therefore, developing college students’ family communication skills is vital in enhancing their support for online learning. Therefore, attention should be paid to cultivating college students’ family communication skills. Schools can lead college students to healthy dialogues with their family members through psychological counseling, special counseling, and teacher–student exchanges; the government and educational institutions should adopt policies or programs of home–school cooperation to improve the interaction between parents and schoolteachers and enhance the understanding of each other, to promote the communication between parents and children better. All these measures will eventually have a positive impact on increasing the interest and effectiveness of online education among college students.

At the external level, social support strongly and positively influenced family communication at the family level, weakly and negatively influenced depression and loneliness at the individual level, and was also a predictor of DSOL among college students. First, the relationship between social support and family communication, depression, loneliness, and DSOL is consistent with the study hypothesis. It may suggest that family communication can also be disturbed by external factors affecting family members’ psychological status and attitudes (DSOL). College students’ social support networks have a significant impact on their academic performance; specifically, support from teachers, friends, and family will ultimately be effective in increasing college students’ academic engagement through their interactions with their family members [70]. Second, the study supports the influence of social support on individuals’ negative psychological status, as suggested in Stress Buffer Theory. Studies have shown that college students feel significantly more stressed after the emergence of the COVID-19 pandemic because they need to cope with other life matters in addition to their learning [71]. When coping with the negative effects of a pandemic, American college students turn first to support from their community, friends, and family to alleviate the stress they face [72]. Therefore, social support has a buffering effect on the negative psychological state of college students. Third, the present study confirms that social support affects DSOL among college students via family communication, depression, and loneliness. Therefore, educational institutions should recognize the importance of creating good learning conditions for college students: emphasizing the role of teachers in guiding students and enhancing peer-to-peer communication opportunities, to provide a complete social support network for college students, alleviating their possible psychological problems and increasing their participation in online learning.

In conclusion, the state of online learning among college students will continue due to the strict prevention and control of the pandemic in China. In order to increase college students’ motivation for online learning, government and education institutions should focus on college students’ mental health issues and consider family interventions and general support that college students require.

### 4.1. Values of the Study

The value of this study includes: in terms of theoretical implications: (1) family has an important influence on students’ learning, and previous studies have examined this relationship more from the overall perspective of family, but this paper studies the relationship between family communication and college students’ learning conditions from a micro-perspective, which broadens the depth of the relationship between family and college students’ learning and has theoretical values; (2) this paper systematically constructs an analytical model from the psychological, family, and social levels that affect college students’ attitudes toward online learning, which has theoretical implications. In terms of practical implications: (1) based on the background of the COVID-19 pandemic in China, the extent of college students’ support for online learning is investigated, which has a practical meaning for understanding the learning status of college students; the research results show that most Chinese college students have positive attitudes toward online learning, which has reference value for the government and educational institutions to understand students’ attitudes toward online learning; (2) this paper explores the influence of college students’ DSOL from the personal and family perspective. The findings suggest that the government and educational institutions should pay more attention to the mental health of college students, and can adopt psychological counseling and provide assistance to alleviate the negative psychological status of college students in the background of the pandemic; in addition, family intervention is also an important means of intervention, and the government and educational institutions should focus on establishing a close relationship with college students’ families. The participation of college students in online learning can be increased through home–school communication, home–school cooperation, and parental education.

### 4.2. Limitations

There were still some limitations that need to be improved in future studies. First, self-report questionnaires were used in our study; therefore, the results of the questionnaire may be affected by participants’ subjective experiences. Second, psychological factors need to be sufficiently considered, such as fear of viral infection in the background of the COVID-19 pandemic, and this deserves to be explored in depth in future studies. Future research could further distinguish the effects of mental health statuses on online learning among college students. Third, structural equation modeling cannot reveal causal effects, and follow-up studies can be conducted longitudinally. Fourth, in this paper, our data were collected during the summer, when participants responded to the questionnaire by remembering their experiences with online learning, leading to some bias and needing to be considered in subsequent studies. Finally, the findings of this paper cannot fully represent the online learning experience of all college students and, therefore, can only reflect certain issues from the side, an issue that needs attention.

## 5. Conclusions

This study investigated college students’ degree of support for online learning during the pandemic. It revealed the influencing factors at three levels: personal, family, and external, by constructing structural equation models. The results indicate that negative psychological status, such as depression, decreases support for online learning, while social support and family communication can provide mitigation. Therefore, it is necessary to increase college students’ online learning participation by providing support and assistance from schools, teachers, society, and families.

## Figures and Tables

**Figure 1 ijerph-19-16814-f001:**
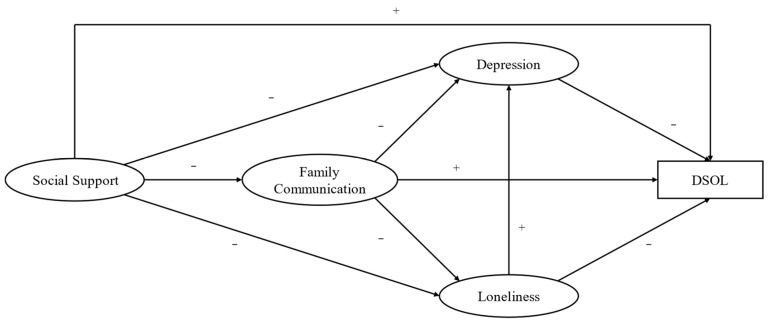
Hypothetical model. Note: “−”means the negative associations; “+” means the positive associations.

**Figure 2 ijerph-19-16814-f002:**
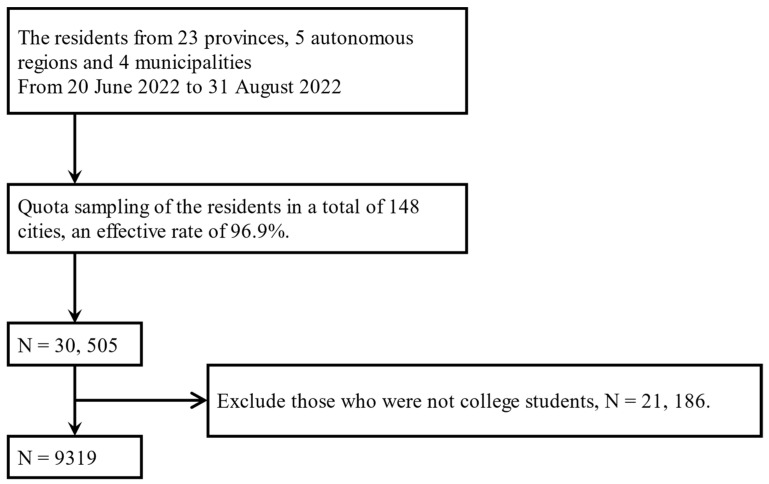
Flowchart of participant enrollment.

**Figure 3 ijerph-19-16814-f003:**
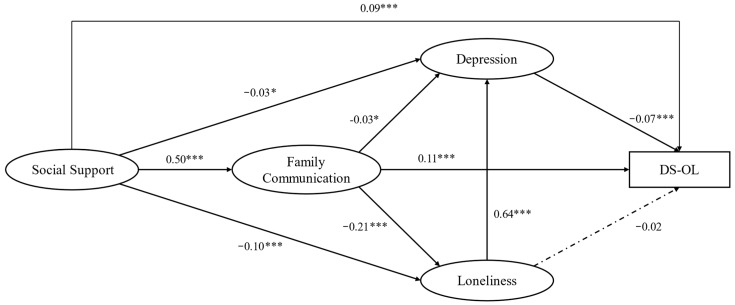
The structural equation model for mediations between social support and DSOL. Note: * *p* < 0.05; *** *p* < 0.001.

**Table 1 ijerph-19-16814-t001:** Descriptive statistics of the participants (*n* = 9319).

Variables	M ± SD or *n* (%)
Age	20.55 ± 4.02
Sex	
Male	3747 (40.2)
Female	5572 (59.8)
Location of residence	
Town	4699 (50.4)
Rural area	4629 (49.6)
Self-reported quarantine status	
Quarantined at home	320 (3.4)
Community under closed management	408 (4.4)
City under closed management	570 (6.1)
Monthly per capita family income	
≤1000	759 (8.1)
1001–2000	1133 (12.2)
2001–3000	1356 (14.6)
3001–4000	1293 (13.9)
4001–5000	1108 (11.9)
5001–6000	966 (10.4)
6001–9000	1086 (11.7)
9001–12,000	709 (7.6)
12,001–15,000	385 (4.1)
≥15,000	524 (5.6)
Degree of support for online learning	66.42 ± 28.73
Not supportive	1312 (14.1)
General	3219 (34.5)
Supportive	4788 (51.4)
Depression	16.70 ± 5.74
Loneliness	4.97 ± 1.68
Family Communication	36.60 ± 8.94
Social Support	14.95 ± 3.85

**Table 2 ijerph-19-16814-t002:** Comparison of the degree of support of online learning among gender, location of residence, and self-reported quarantine status.

Variables	*n* (%)			*χ^2^*	*p* Value
	Not Supportive	General	Supportive		
Gender					
Male	585 (44.6)	1221 (37.9)	1941 (40.5)	17.631 ***	<0.001
Female	727 (55.4)	1998 (62.1)	2847 (59.5)		
Location of residence					
Town	665 (50.7)	1595 (49.5)	2439 (50.9)	1.530	0.465
Rural area	647 (49.3)	1624 (50.5)	2349 (49.1)		
Self-reported quarantined status					
Quarantined at home	72 (5.5)	99 (3.1)	149 (3.1)	19.435 ***	<0.001
Not quarantined at home	1240 (94.5)	3120 (96.9)	4639 (96.9)		
Community under closed management	74 (5.6)	132 (4.1)	202 (4.2)	5.874	0.053
Community under no closed management	1238 (94.4)	3087 (95.9)	4586 (95.8)		
City under closed management	100 (7.6)	197 (6.1)	273 (5.7)	6.612 *	<0.05
City under no closed management	1212 (92.4)	3022 (93.9)	4515 (94.3)		

Note: * *p* < 0.05; *** *p* < 0.001.

**Table 3 ijerph-19-16814-t003:** Associations between social support, family communication, depression, loneliness, and the degree of support of online learning (*n* = 9319).

Correlation Coefficients	Social Support	Family Communication	Depression	Loneliness
Family Communication	0.467 **	1		
Depression	−0.156 **	−0.197 **	1	
Loneliness	−0.183 **	−0.244 **	0.580 **	1
DSOL	0.154 **	0.173 **	−0.118 **	−0.107 **

Note: DSOL = Degree of support for online learning, ** *p* < 0.01.

**Table 4 ijerph-19-16814-t004:** The results of the estimations.

Hypotheses		Parameter Estimations	
		Coefficient	Is the Hypothesis Supported?
H1a	DE-DSOL	−0.07 ***	YES
H1b	LL-DSOL	−0.02	NO
H2	FC-DSOL	0.11 ***	YES
H3	SS-DSOL	0.09 ***	YES
H4a	SS-DE	−0.03 *	YES
H4b	SS-LL	−0.10 ***	YES
H5a	FC-DE	−0.03 ***	YES
H5b	FC-LL	−0.21 ***	YES
H6	SS-FC	0.50 ***	YES

Notes: * represents statistically significant at 5% and *** represents statistically significant at 1‰. SS = Social Support; DE = Depression; LL = Loneliness; FC = Family Communication; DSOL = Degree of support for online learning.

**Table 5 ijerph-19-16814-t005:** Path indicators and proportions of mediating effects between social support and the degree of support of online learning.

Paths	S.C.	T.E.	D.E.	I.E.	S.E.	P.E.	Biased-Corrected 95% CI	
							Lower Limit	Upper Limit
SS-FC-DSOL	0.055	0.145	0.090 ***	0.055	0.007	37.93%	0.042	0.069
SS-DE-DSOL	0.002	0.092	0.090 ***	0.002	0.001	2.17%	0.000	0.004
SS-FC-DE-DSOL	0.001	0.091	0.090 ***	0.001	0.000	1.10%	0.000	0.002
SS-LL-DE-DSOL	0.005	0.095	0.090 ***	0.005	0.001	5.26%	0.002	0.007
SS-FC-LL-DE-DSOL	0.005	0.095	0.090 ***	0.005	0.001	5.26%	0.003	0.007

Notes: Only an indirect path with an empirical 95% confidence interval is presented, and it does not overlap with zero; bootstrap sample size = 2000; *** represents statistically significant at 1‰ level. S.C. = Standardized coefficient; S.E. = Standard errors; T.E. = Total effects; D.E. = Direct effects; I.E. = Indirect effects; P.E. = Proportional values of indirect-only-mediated effects; SS = Social Support; DE = Depression; LL = Loneliness; FC = Family Communication; DSOL = Degree of support for online learning.

## Data Availability

Data are available, upon reasonable request, by emailing: bjmuwuyibo@outlook.com.

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
