# Peer review of "College Students’ Degree of Support for Online Learning during the COVID-19 Pandemic and Associated Factors: A Cross-Sectional Study"

_ijerph, 2022, doi:10.3390/ijerph192416814_

Round 1

Reviewer 1 Report

The paper needs close and serious editing. More information on how the results of social support, depression etc were arrived at is needed. The implications of the results needs more elaboration. How do the results of your study help in online education? How can other educational systems benefit from the results? In general the implications of the results have not been properly discussed. 

Reviewer 2 Report

Dear authors,

This is an interesting scientific article, however it is suggested to add relevant research from the last five years on social environment support and family communication can contribute to improve the psychological state and learning of university students. In this sense, it is advisable to consult the Web of Science and SCOPUS databases.

It is also advisable to develop a greater analysis of the conclusions obtained.

It is recommended that the usefulness and contribution of the study to the scientific community be explored in greater depth.

Congratulations once again to the authors and I hope that my contributions can help to improve the document.

Reviewer 3 Report

1. Since the survey was conducted mainly in the summer months of July and August, there are not much online courses, therefore the limitation is obvious, the results are not so representative.

2. The online study on dormitory during the campus management in the school time, was not included, so the survey is not comprehensive.

3. The fealful of virus  infection was not considered in psychiatric problem.  

Reviewer 4 Report

Dear authors, I have added my comments in the attached file! Best wishes!

Round 2

Reviewer 4 Report

Thank you very much for the improved article in accordance with all the recommendations. Good luck!